# ERG and Behavioral CFF in Light-Damaged Albino Rats

**DOI:** 10.3390/ijms23084127

**Published:** 2022-04-08

**Authors:** Glen R. Rubin, Yuquan Wen, Michael S. Loop, Timothy W. Kraft

**Affiliations:** 1Department of Optometry and Vision Sciences, University of Alabama at Birmingham, 1720 2nd Avenue South, Birmingham, AL 35294, USA; rubinglen@gmail.com (G.R.R.); loop2211@gmail.com (M.S.L.); 2Department of Neurobiology, University of Alabama at Birmingham, 1720 2nd Avenue South, Birmingham, AL 35294, USA; yuquan.wen@outlook.com

**Keywords:** flicker, CFF, light damage, ERG, behavior, scotopic, photopic, a-wave, b-wave, degeneration

## Abstract

The full-field ERG is useful for index rod- or cone-mediated retinal function in rodent models of retinal degeneration. However, the relationship between the ERG response amplitudes and visually guided behavior, such as flicker detection, is not well understood. A comparison of ERG to behavioral responses in a light-damage model of retinal degeneration allows us to better understand the functional implications of electrophysiological changes. Flicker-ERG and behavioral responses to flicker were used to determine critical flicker frequency (CFF) under scotopic and photopic conditions before and up to 90 d after a 10-day period of low-intensity light damage. Dark- and light-adapted ERG flash responses were significantly reduced after light damage. The a-wave was permanently reduced, while the b-wave amplitude recovered over three weeks after light damage. There was a small, but significant dip in scotopic ERG CFF. Photopic behavioral CFF was slightly lower following light damage. The recovery of the b-wave amplitude and flicker sensitivity demonstrates the plasticity of retinal circuits following photopic injury.

## 1. Introduction

The full-field electroretinogram (ERG) is commonly used to quantify retinal function in humans [1] as well as rodent models [2] of retinal degeneration. Abnormal ERG recordings can assist in the early detection of photoreceptor degenerations such as retinitis pigmentosa (RP). Degeneration is characterized by initial rod apoptosis followed by cone loss, suggesting a dependence of cone survival upon rods [3,4]. Cone-mediated ERGs are often used clinically to monitor disease progression, due to an early loss of the rod signal [5]. The rod-mediated ERG signal may become more clinically relevant if therapeutic advances [3,6,7] and/or retinal plasticity [8] can repair rod vision. Regardless, it is not clear whether improvements to an ERG signal, such as a larger a-wave, b-wave, or flicker response, can be directly correlated to improvement in visual function.

Continuous exposure to low-intensity light causes photoreceptor apoptosis in albino animals [9,10,11]. Using this type of light damage as a model for RP has advantages; there are no confounding genetic factors and the extent of degeneration is controllable. LaVail et al., demonstrated the sparing of cones over rods in albino rats exposed to 700 lux for 54 days [12]. Cones represented 60% of the total photoreceptors after 178 days of exposure. Cicerone and colleagues [13,14] found evidence that the photopic system was relatively spared after continuous exposure to light. ERG measures of dark adaptation in albino rats exposed to 1080 lux for up to 24 h showed a significantly elevated rod branch, whereas the cone branch of the adaptation curve was unaltered.

However, the characterization of rods as more vulnerable to light damage is not consistently reported. Sugawara, Sieving and Bush showed reductions in the dark- and light-adapted ERG in rats exposed to 1000–3000 lux for up to 48 h [15]. Sensitivity to a flickering light can be quantitatively assessed by a measure of the critical flicker frequency (CFF), that is the maximum flicker frequency that will elicit a behavioral response or a criterion voltage in an ERG recording. Williams et al., observed deficits in behavioral CFF in rats exposed to 500 lux for 8 days [16]. The CFF curve was suppressed over an intensity range of six log units, with the largest decrease in CFF in the photopic range. Recently, Riccitelli et al., thoroughly examined the light-damage effects of brief (12–24 h) 1000 lux exposures on retinal morphology, anatomy, function and some molecular signatures as well [17]. They found that functional losses were detectable prior to anatomical changes; and that a loss of 75% of the cones resulted in an immediate loss of photopic b-waves of similar magnitude, but that a substantial recovery was observed, albeit incomplete and transient on the scale of months.

In the present study, rod and cone function was monitored for 90 days after 10 days of exposure to 280-lux fluorescent light. ERG responses were measured by dark- and light-adapted flashes and sinusoidal flicker. Psychophysical measures of CFF were conducted in a water maze. The differences between the effect of light-induced retinal degeneration on threshold measures versus maximum responses were observed. Strong parallels were noted between behavior and ERG measures of flicker threshold. This light-damage model demonstrates significant recovery of the flash ERG, however, CFF thresholds were minimally disturbed by a significant loss of photoreceptors.

## 2. Results

Light damage (LD) treatment led to ERG amplitude reduction. Dark-adapted maximum responses (R_max)_ of the a- and b-wave of the flash response were lower across all experimental days following light damage. Figure 1 shows the typical ERG results and analysis; the representative ERG responses are dramatically reduced immediately following the 10-day light-damage regimen, but after 29 days of recovery, the a-wave is partially restored and the b-wave has substantially recovered. Figure 2 and Table 1 give the mean a- and b-wave R_max_ values measured before light damage and on recovery day (R) 6, 20, 48, and 90 in the groups of control (filled circles) and test animals (open squares). In Figure 2, each animal’s response was normalized to its pre-light damage value. On R6, a- and b-wave R_max_ responses to light-damaged animals (*n* = 9) were a third lower than the pre-light damage responses (*p* < 0.05). By R20, the b-wave recovered over 80% of its original amplitude, whereas the loss of the a-wave was permanent (*p* < 0.05). Table 1 compares control and light-damaged dark-adapted a- and b-wave maximum response amplitudes. The light-damaged a-wave measured significantly lower on R6 and R48 (*p* < 0.001). However, there was no significant difference between the a-wave amplitudes of the light-damage and control groups at R90. Interestingly, b-wave amplitudes were affected very little by the loss of a-waves, there was no decrement in control animals and in addition, the 33% loss of a-waves in the LD group results in only 19% decrement of the b-waves in these animals at R90. 

Cone ERG responses to bright flashes were measured on a 505 nm rod-saturating background (Figure 2c,d). R_max_ values of the light-adapted a- and b-waves were stable in control animals. However, the light-damage group exhibited a progressively declining a-wave. On R6, the light-damage group’s a-wave measured 17% less than pre-light damage (Pre-LD) and steadily declined thereafter. By R90, the a-wave had declined by 36% compared to the Pre-LD values. In contrast, the b-wave showed an initial decline of 30% but thereafter recovered, resulting in only a 10–15% loss of amplitude.

An intensity-response function defined sensitivity (I_1/2,_ photons/µm^2^ at cornea) of the a- and b-waves. Figure 3 shows the calculated I_1/2_ of a- and b-waves at Pre-LD, R6, R20, and R34. At R6, a half-maximal a-wave could be elicited with a 1.7-fold less light than pre-light damage (*p* < 0.05). However, the a-wave sensitivity was not significantly different at R20 and thereafter. Thus, after following the first cohort of animals for 90 days post light damage, the second litter of animals was only followed for 34 days after the end of the light-damage regimen (Figure 3). The relationship between light damage and b-wave sensitivity was less clear. Although I_1/2_ values of the b-wave varied over the experimental period, there were no significant differences compared to Pre-LD I_1/2_ b-wave values. 

Figure 4 shows ERG and behavioral measures of CFF at five time-points: Pre-LD, R6, R20 R48, and R90. The ERG CFF values (Figure 4c,d) were generally higher than behavioral CFFs (Figure 4a,b) under both scotopic and photopic conditions. In control animals, scotopic ERG CFF measured 24.7 Hz, while behavioral CFF measured 20.3 Hz at the start of our experiments. A comparison of ERG CFF to behavioral CFF in control animals showed ERG CFF to be statistically higher (*p* = 0.002) than behavioral CFF under scotopic conditions. Light-damaged animals did not exhibit significant behavioral differences under scotopic conditions and for ERG measures, only results from day R6 showed significant scotopic CFF loss (*p* < 0.01).

Under photopic conditions, the initial ERG CFF was higher (47.8 Hz) than behavioral CFF (40.8 Hz). Behavioral CFF for photopic lighting was significantly depressed (*p* < 0.001) over the entire recovery period between R6 and R48 (Figure 4a); while the ERG measures were stable and showed no detectable difference between control vs. light-damaged animals (Figure 4c). In control animals, there was no significant difference between ERG CFF and behavioral CFF beyond R20 under either scotopic or photopic conditions. Comparing the two techniques of measurement, light-damaged animals showed lower behavioral CFF (Figure 4a,b open squares) when compared to the ERG-based measures of CFF (Figure 4c,d open squares) in both scotopic (*p* < 0.05) and photopic (*p* < 0.01) conditions. Thus, the signals we measure with flicker-ERG do not match the behavioral measures of flicker sensitivity in our forced-choice swimming test. 

Photoreceptor loss was quantified by comparing the outer (ONL) and inner (INL) nuclear layer thickness of light-damaged versus control animals (Figure 5 and Figure 6). The outer/inner segment (OS/IS) layer, as well as the outer nuclear layer thickness, were significantly less than normal for control animals. In Figure 6, ONL and INL thickness are plotted as a function of distance from the center of the optic nerve head (ONH) for control (*n* = 3; filled circles) and light-damaged retinas (*n* = 7; open squares). The calculated area of the ONL for a section of the retina from light-damaged animals was 324 µm^2^, significantly less (*p* < 0.01) than the matched area of ONL in the control animals (417 µM^2^). INL areas were also compared in light-damaged animals to controls, however, the difference was not statistically significant. The measures of ONL thickness matched the same comparison made by ONL nuclei counts (Appendix A).

## 3. Discussion

Albino rats exposed to low-intensity light for 10 days showed transient changes in b-wave amplitudes, a-wave sensitivity, and CFF values. These changes were markedly different from the results obtained previously, detailing the permanent loss of the rod- and cone-driven ERG flicker signal in aging RCS rats [18] and more severe forms of light damage in albino animals [17,19,20]. We found a uniform reduction in the ONL from the superior to the inferior retina, whereas others have noted focal damage under their conditions. It is possible that the lower light levels (280 vs. 1000×) and longer durations (1 vs. 10 days) permitted animal movement over the 10-day period that was either less inhibited by the light levels, or that normal feeding/socialization behavior resulted in more uniform illumination of the retina and thus, more uniform damage. However, the non-homogeneous loss of cones in this model [21] and the known regionalization of damage and repair in other studies suggest a layering of complex physiological activity and reactivity in retinal degenerations [20,22]. 

In the present study, we measured a 33% loss of dark-adapted b-wave amplitude six days after light damage. At the end of our study, ONL thickness showed a persistent 22% loss, but the b-wave had recovered. Other investigators have noted b-wave recovery after photoreceptor loss [17,23,24] and retinal remodeling is known to occur following photoreceptor loss [25,26,27], which may explain the recovery of b-wave amplitudes. Considering the similar loss of the b-wave amplitude in the RCS (PN23) and light-damaged albino rats (R6), we expected to see greater reductions in ERG CFF in the light-damaged rats. Scotopic ERG CFF was reduced by 11% after light damage, while there were no changes in photopic ERG CFF. In RCS, degeneration stems from the inability of RPE cells to properly phagocytose shed rod outer segments owing to a defect in the Mertk tyrosine kinase receptor gene [28]. The build-up of these shed disks in the subretinal space leads to the formation of a debris zone and ensuing photoreceptor apoptosis [29]. The exact mechanism of cell death is unclear; the debris zone may interfere with the diffusion of metabolites [30] or oxygen [31]. The lower CFF in RCS rats may be due to the retina’s chronic unmet metabolic demand or some associated degenerative stresses. In contrast, photoreceptor apoptosis caused by light damage [32] is the result of a temporary insult. The threshold flicker function may be robust in the face of this transitory stress, so as to fully recover six days after the exposure period. While the retina in the RCS rats has no time to recover, a plastic adjustment in gain along the rod-bipolar pathway in the light-damaged retina may partially compensate and maintain b-wave amplitudes. 

We previously tested the flicker function in a ß subunit knockout (KO) mouse model (*Cngb1-x1*) [33]. In this model of RP, CFF curves were suppressed only over the scotopic range. Similarly, human testing has shown an earlier loss of the rod-driven flicker signal over photopic CFF in RP patients [34]. Further flicker ERG testing in rodent rescue models, which exhibit decrements to CFF, would help to clarify how flicker function is related to the degenerative condition. 

A 3 µV criterion of response was used for determining ERG CFF since it is near the root mean squared noise in our averaged recordings. Other studies have also used a 3 µV criterion amplitude for flicker ERG analysis [18,35]. Initially, it was unclear how this ERG criterion would correspond to the behavioral CFF. Behavioral CFF was calculated as 75% correct in our two-alternative forced-choice testing. Overall, there was a fair correspondence between ERG and behaviorally determined CFF values. The scotopic ERG CFF was significantly lower (24.7 vs. 20.3 Hz) in control animals. Light-damaged animals exhibited significant differences between the ERG criterion and behavioral CFF under both scotopic and photopic conditions. This interesting result suggests that, after light damage, a larger ERG signal is required in order to mediate the behavioral detection of a threshold flicker signal for cone-driven visual function; perhaps indicating a temporary perturbation in cone response effectiveness, as seen by ganglion cells. We also tested an alternative ERG criterion of 5 µV in order to find an electrophysiological criterion that better represented the psychophysical threshold. Using the 5 µV criterion eliminated any statistical difference between ERG and behavioral measures of CFF in both control and light-damaged animals. 

A single study by Coile et al., reported behavioral CFF exceeding ERG CFF measures in dogs [36]. However, our results agree with the majority of previous studies in primates showing ERG measures of CFF to be comparable to or higher than those obtained psychophysically [37,38,39], suggesting that the visual system is sensitive to nearly threshold temporal responses in the retina.

There was a significant dip in scotopic ERG CFF after light damage at R6. This temporary deficit may be the result of photostasis [40], in which the length of the rod’s outer segments shorten. We did not find reductions in behavioral CFF due to light damage of the same magnitude described by Williams et al. [16]. Williams et al., tested CFF in albino rats exposed for eight days to a relatively higher intensity light (500 lux), resulting in a loss of 90% ONL thickness. Our model of partial damage (22% loss) did not exhibit significant behavioral or ERG deficits in flicker sensitivity (CFF). Thus, the threshold for significant and permanent behavioral deficits lies in between these damage levels. In our experiments, the lighting in the standard rat housing was higher than that of earlier studies, and thus the damaging light/standard light ratio was much lower in our experiments. It has been previously noted that exposures to periodic bright lights in advance of damaging offers some protection [41], and thus in future studies, the general housing lighting conditions should be below 50 lux and ideally even below 20 lux.

After light damage, there was a small decrease in the light-adapted a-wave at R6, suggesting an impairment of cone function. Over the time course of these trials, the impairment progressively worsened, such that at R90, a statistically significant difference in the a-wave amplitude was noted. This raises the interesting possibility that cone receptors continue to degenerate after light damage, which may also help explain the apparent greater disparity between the ERG and behavioral criterion. Several studies have documented long-term apoptosis in ONL after exposure to high-intensity light [10,42,43,44,45] The b-wave and flicker responses may not reflect this loss due to greater plasticity within the b-wave generating circuitry. Indeed, others have noted upregulation in neuroprotective factors after light damage [17,46]. In a companion study to that presented here, Benthal et al., found that 10 days of low-level light damage results in the loss of 15% of the central cone population while sparing the lower density of peripheral cones [21]. Thus, the short term and long-term loss of cones need to be carefully documented, even in situations such as low-level light damage where cone resilience is expected.

Our study showed that CFF was less affected by light damage than the ERG b-wave amplitude. The minor changes to CFF demonstrate the preservation of threshold responses that mediate behavior despite significant photoreceptor degeneration. The CFF was not significantly different from control animals at the end of our study. These findings suggest that a level of retinal resilience preserves functional circuits or computations despite the loss of photoreceptors. Because the damage induced by low-level constant light damage is easily controlled, it affords the experimenter the opportunity to test the protective measures given in advance of damage, concurrent with the ongoing physiologic insult or during the recovery period. It is known that retinal damage can induce upregulation of neuroprotective mechanisms and compounds and, that enhancing those reactions, for example, by photobiomodulation, can help to preserve retinal structure and function [22,47]. In our hands, the behavioral tests are cumbersome and time-consuming, but the ERG tests of CFF, a-wave and b-wave maxima are both effective and efficient and can reliably report the loss or preservation of retinal function in the face of environmental or genetically induced degeneration.

## 4. Materials and Methods

### 4.1. Animals

Sprague Dawley albino rats were housed under 12-h light/dark conditions (57–140 lux). Animals were trained on a behavioral task for 65 days prior to light damage. All animals were handled according to the principles of the ARVO Statement for the Use of Animals in Ophthalmic and Vision Research.

### 4.2. Electroretinography

Details of the ERG recording follow those of Rubin and Kraft [18]. Briefly, recordings were made prior to light damage and regularly during the 90-day recovery period. ERGs were always recorded after behavioral testing to eliminate any possible lingering effects of anesthesia during the swim tests. Rats were dark-adapted overnight, corneas were anesthetized and pupils dilated. The light source was a 100-W tungsten-halogen lamp focused on one end of a fiber optic. Full-field ERGs were recorded using a 2 mm diameter platinum wire loop embedded in the tapered end of a hollow Plexiglas rod [48]. The tapered end acted as a diffusing element, yielding an isotropic plane of illumination. The reference electrode was a second platinum loop placed on the non-stimulated eye.

Stimulus intensity was controlled by calibrated neutral-density filters, and the wavelength was 505 nm (Andover Co., Salem, NH, USA; 37 nm bandwidth). ERG responses were obtained using 3 to 20 repeats of a 10 ms stimulus [49]. The inter-stimulus interval (ISI) ranged from 2.2 s up to 30 s. An intensity-response function was generated for both a- and b-waves. A bright camera flash, filtered by a 530 nm (10 nm bandwidth) interference filter, was used to evoke maximum responses. Light-adapted ERGs to the same bright flash were recorded in the presence of a 505 nm rod-saturating adapting field (3.66 × 10^4^ photons µm^−2^ s^−1^) incident upon the cornea. The wavelength of the stimuli was chosen to match or be near the optimum wavelength for rat rhodopsin and cone opsin [50]. Under dark-adapted conditions, these bright flashes were delivered at an ISI of 120 s; under light-adapted conditions, ISI was 60 s. ERG a-wave amplitudes were measured from the pre-flash baseline, and b-wave amplitudes were measured from the trough of the a-wave (when present) to the peak of the b-wave.

Sinusoidal flicker was produced by a ferro-electric liquid crystal shutter (LV050; Displaytech, Boulder, CO, USA) driven by a pulse-width modulation paradigm. The average Michelson contrast of the flicker stimulus was 0.86. The stimulus frequencies were: 1, 2, 4, 5, 10, 16, 20, 25, 32, and 40 Hz. The on-transient of the 5 s response was ignored and the final 4.5 s of data were averaged into a one- or two-cycle wave to measure the response amplitude. The log_10_ of the response amplitudes for each intensity was plotted and fitted with a line to determine electrophysiological CFF using a 3µV criterion voltage [35].

### 4.3. Behavior

Behavioral CFF was determined by two alternative forced-choice tests in a water maze [51,52]. A trial began when an animal was released at the narrow end of a gray acrylic trapezoidal-shaped pool of water (~22 °C, 15 cm deep) (Appendix A). At the opposite wider end of the pool were two side-by-side displays positioned just above the water. A centered divider (40 × 40 cm) separated the two displays. Typically, the rats swam the length of the maze (140 cm) in 3 s. A display would be either flickering or steady. A hidden platform was placed under the steady display.

The two displays were constructed by placing a translucent sheet (ca 17 cm square) in front of a bank of green LEDs enclosed in a mylar coated funnel. The opening at the end of the funnel (1 ¾ cm) sat flush against the acrylic sheet. Inserting neutral-density filters between the funnel and acrylic sheet attenuated the light intensity. A specially fabricated circuit drove one display at an adjustable flickering frequency (5–80 Hz) and the other at a steady fixed high frequency (180 Hz). The position (left/right) of the steady vs. flickering displays could be switched, and the two displays were matched in time-averaged luminance.

Behavioral testing was accomplished in 3 phases [52]. Initially, the rat was introduced into the pool a few inches in front of the steady (correct) stimulus and platform. After finding the platform, the rat was allowed to remain on it for a few seconds before being returned to its holding cage. The position of the steady (correct) stimulus and platform was moved from side to side. The starting distance was increased when the rat reliably swam directly to the platform. A schematic diagram of the testing apparatus is given in Appendix A.

Next, the rat learned to discriminate between the steady and flickering (~6 Hz) stimuli. The rat was sequestered behind a clear starting gate and then required to locate the platform in front of the correct stimulus before being removed from the maze. If the rat passed the divider towards the flickering stimulus, the trial was recorded as an error. This training phase continued until the rat consistently (+80%, ~40 trials) selected the steady stimulus.

CFF was determined by increasing the temporal frequency following a method of limits staircase procedure [53]. In the first trial, the rat had to discriminate between 6 Hz and the steady stimulus; if the correct frequency was raised in 5 Hz increments. If two consecutive incorrect responses were made, the frequency was lowered by 10 Hz. Thereafter, every consecutive incorrect discrimination resulted in lowering the frequency by 5 Hz. Upon correct discrimination, the stimulus frequency was increased (5 Hz) until two consecutive incorrect responses. This procedure continued until 3–5 reversals were made and the threshold was taken as 75% correct.

The position of the correct stimulus during CFF testing followed a Gellerman series [54]. If an animal went to the same side on four consecutive trials, the correct stimulus was retained on the non-preferred side until a correct response was made. Testing sessions were ~25 trials, with up to 2 sessions per day. A CFF frequency-of-seeing curve was generated in 2 to 3 days based on about 80 trials. Two luminance levels were tested (scotopic 6.50 × 10^3^ µm^−2^ s^−1^ photons; photopic 6.38 × 10^6^ µm^−2^ s^−1^ photons).

### 4.4. Light Damage

Transparent cages were split along the long axis with one rat per compartment. Minimum bedding was used in the cages. Water and food were provided ad libitum. A fluorescent light bulb (T8, (Phillips, Elgin, IL, USA) 48 inches, 32 W, 2950 lumens, color rendering 85, correlated color temperature 3000 K) was placed above the animal cages for an average illuminance of 280 + 20 lux (Tektronix J-16/J6511, Richardson, TX, USA). Following the light damage, rats were returned to cyclic (12/12) lighting.

### 4.5. Histology 

Rats were euthanized by CO_2_ asphyxiation, and whole eyes were harvested and fixed overnight (2.5% glutaraldehyde, 1% paraformaldehyde). The tissue was dehydrated and then embedded in plastic (JB-4 Embedding Kit, Polysciences, Warrington, PA, USA). Radial sections, 5 μM thick, were cut (Leica Microsystems, Wetzlar, Germany) and collected at 500 µM intervals. Slides were stained with Hematoxylin and Eosin and retinal sections were imaged for measures of outer nuclear layer (ONL) and inner nuclear layer (INL) thickness. Measurements were taken at 450 µM increments, starting from the optic nerve head (ONH) in both inferior and superior hemispheres. Only animals surviving 48 days past light, or sham, damage were considered in the statistical analysis of histological data.

### 4.6. Statistical Analysis

The between-group significance for ERG data collected at specific experimental time-points (R6, R20, R48, etc.) was calculated by a 2-tailed 2-sample *t*-test with equal variance assumed. Within-subjects comparisons were made by a paired *t*-test. Flicker analysis and comparisons between the ERG and behavioral CFF were made by repeated-measures ANOVA. Histological analysis was made by a 2-tailed 2-sample *t*-test. Equal variance and normal distributions were assumed.

## Figures and Tables

**Figure 1 ijms-23-04127-f001:**
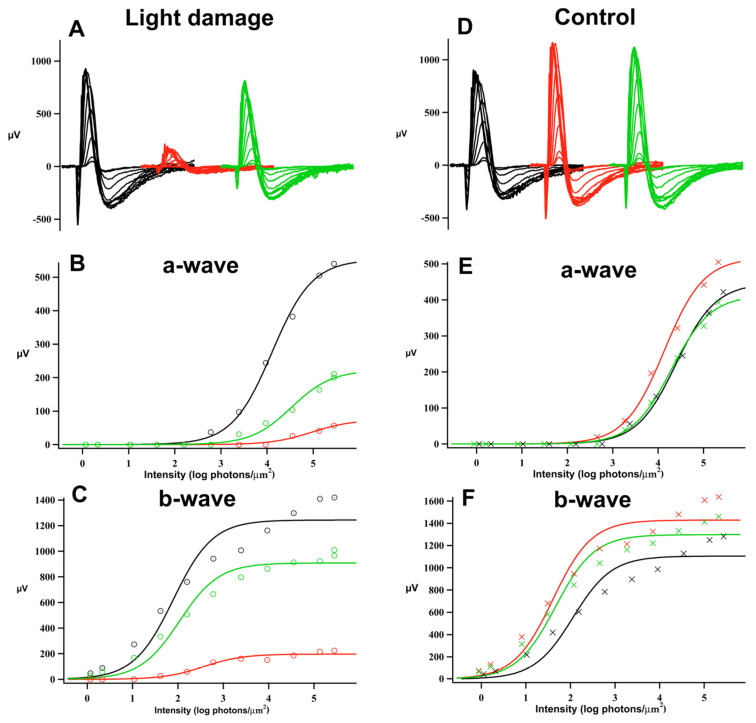
Sample ERG responses and analysis. Functional evaluation of photoreceptor degeneration. (**A**) ERG response families prior to light damage (Pre, black), one day after light damage (R1, red), and 29 days after light damage (R29, green). (**D**) ERG response families prior to sham light damage (Pre, black), one day after sham light damage (C1, red), and 29 days after sham light damage (C29, green). (**B**,**C**) Intensity-response plots for a–wave and b–wave after light damage. (**E**,**F**) Intensity-response plots for a-wave and b-wave after sham light damage. The plots were fitted with a modified Michaelis–Menten function (continuous line). Note the significant reduction of a-wave amplitudes and b-wave amplitudes after light damage and the following recovery (**B**,**C**).

**Figure 2 ijms-23-04127-f002:**
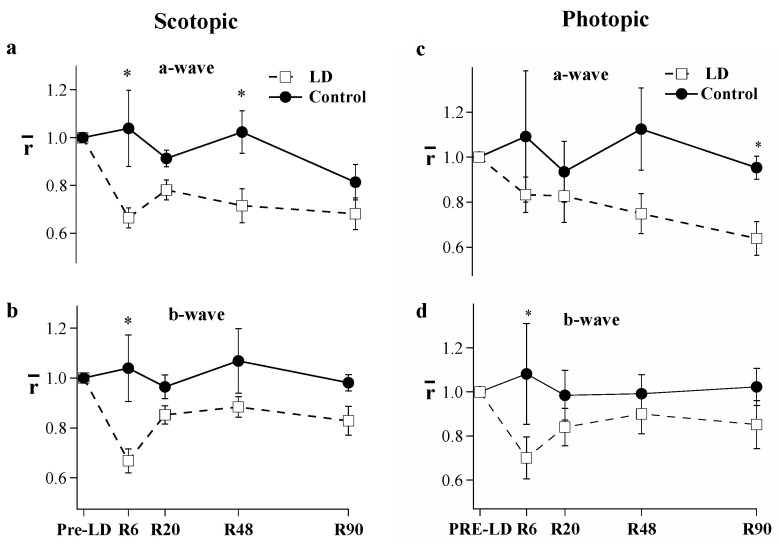
Comparison of the dark-adapted (**a**,**b**) and light-adapted (**c,d**) flash ERG (Rmax) between light-damaged (open squares, *n* = 5–10) and control animals (filled circles *n* = 3–6). Recordings were made before light damage and on recovery day (R) 6, 20, 48, and 90. Each animal’s response was normalized to its pre-light damage value. Error Bars = ± SEM. (* *p* < 0.05).

**Figure 3 ijms-23-04127-f003:**
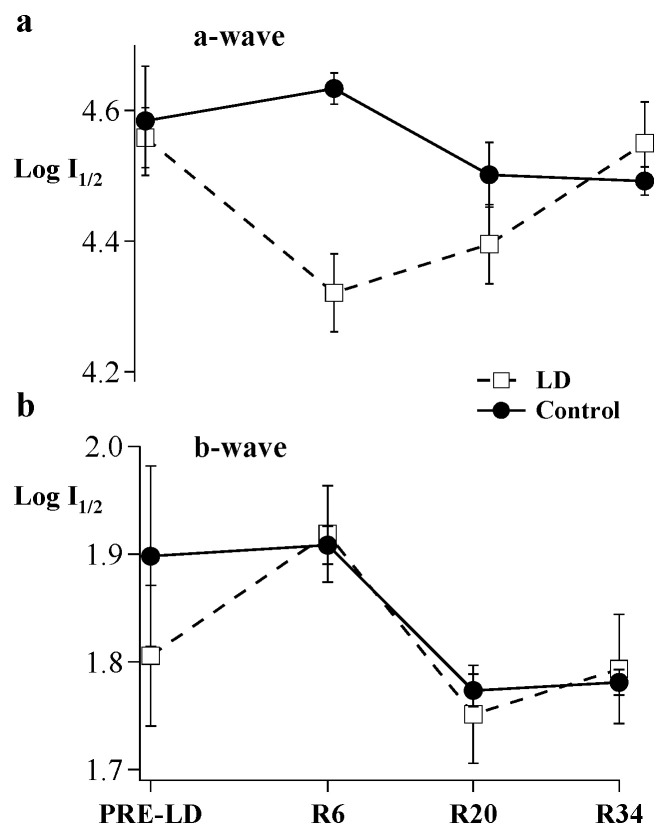
An intensity response (IR) series was recorded by progressively increasing flash intensity. Resulting curves were fit by a modified Michaelis function. Sensitivity (Log I_1/2_) for (**a**) a-and (**b**) b-waves were calculated before light damage and on R6, R20, and R34. Open squares connected by dashed lines show results for light-damaged animals (*n* = 6 to 10). Filled circles connected by solid lines show results for control animals (*n* = 3 to 6), Error Bars = ± SEM.

**Figure 4 ijms-23-04127-f004:**
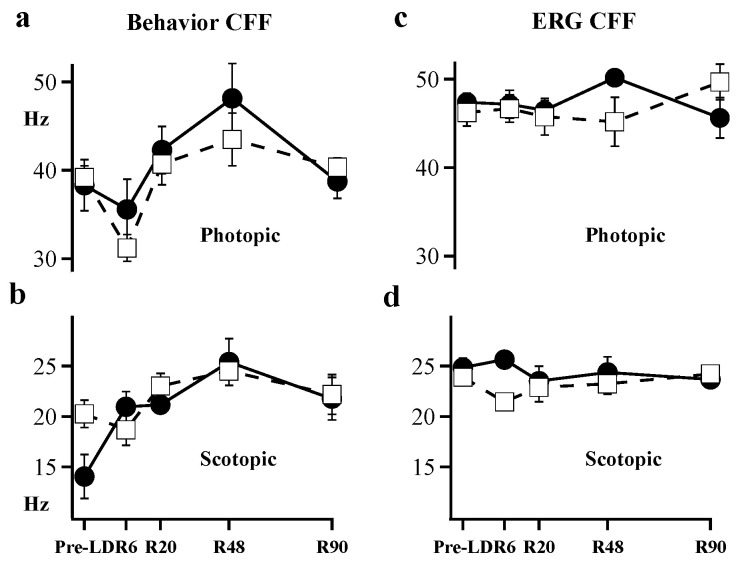
Behavior CFF was measured using (**a**) photopic intensity and (**b**) scotopic intensity. ERG CFF was also measured under (**c**) photopic and (**d**) scotopic intensities. Filled circles connected by solid lines show results for control animals (*n* = 3 to 6). Open squares connected by dashed lines show results for light-damaged animals (*n* = 5 to 10). Error Bars = ± SEM; where absent error bars are within the data marker.

**Figure 5 ijms-23-04127-f005:**
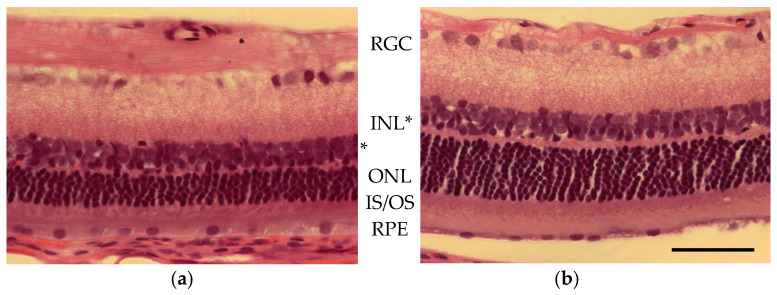
Photographs from a section of retina approximately 1.3 mm inferior to the ONH in (**a**) light-damaged and (**b**) control animal (hematoxylin and eosin stained). Cell body layers labeled from the top of the images; GCL (ganglion cell layer), (INL) inner nuclear layer (indicated with an asterisk * due to changes in retinal thickness), ONL (outer nuclear layer), IS/OS (the inner/outer segment of the photoreceptors), and RPE (retinal pigmented epithelium). Scale bar = 50 µm.

**Figure 6 ijms-23-04127-f006:**
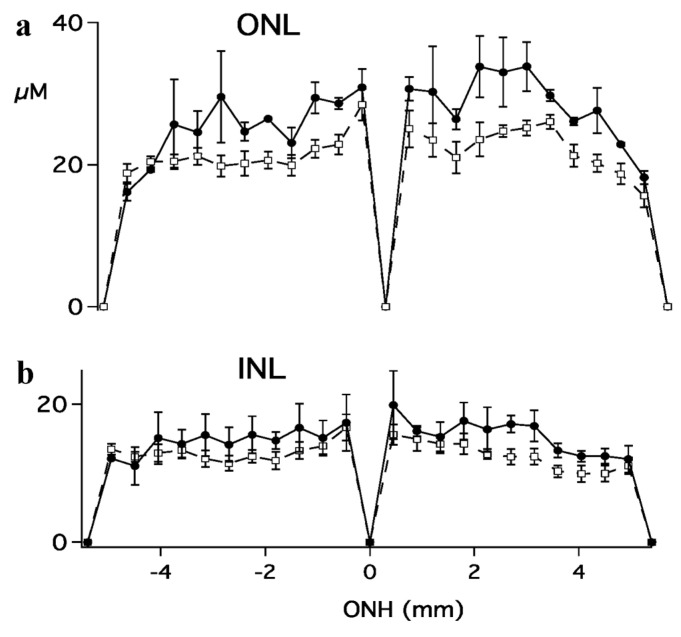
The thickness of (**a**) outer and (**b**) inner nuclear layers plotted as a function of distance from the ONH. Measures were made along a vertical meridian within 2 mm of the ONH. The negative values on the *x*–axis indicate inferior retina, while positive indicate superior retina.

**Table 1 ijms-23-04127-t001:** Comparison of dark-adapted ERG responses (µV) for control and light-damaged animals; Mean ± SEM (*n*). *p* values are comparison of Pre-LD vs. R90 within each column (*n* = 3 WT, *n* = 5 LD).

Animal	a-Wave		b-Wave	
	Control	LD	Control	LD
Pre-LD	367 ± 19 (5)	354 ± 7 (10)	1440 ± 123(5)	1450 ± 78 (10)
R6	363 ± 42 (4)	236 ± 15 (9)	1420 ± 144 (4)	930 ± 64 (9)
R20	313 ± 9 (3)	277 ± 20 (7)	1230 ± 55 (3)	1200 ± 57 (7)
R48	349 ± 9 (3)	250 ± 30 (5)	1350 ± 49 (3)	1250 ± 74 (5)
R90	278 ± 16 (3)	237 ± 5 (5)	1260 ± 126 (3)	1180 ± 95 (5)
	(*p* = 0.15)	(*p* < 0.01)	(*p* = 0.62)	(*p* < 0.05)

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
