# Peer review of "ERG and Behavioral CFF in Light-Damaged Albino Rats"

_ijms, 2022, doi:10.3390/ijms23084127_

Round 1

Reviewer 1 Report

In this manuscript, the authors describe their experiments correlating ERG CFF and behavioral CFF after light-damage. They expose albino rats to 10 days of low light intensity damage and then assess ERGs and behavior 6, 20, 48, and 90 days after exposure. Their results indicate that light-damage reduced ERG amplitude at R6, with deficits in both a-wave and b-wave amplitudes.  A-wave amplitudes displayed a permanent reduction, while b-wave amplitude displayed recovery at the later time points.  The authors also report differences in light-damage effect between ERG CFF and behavioral CFF, with greater changes in ERG responses.  This work is relevant as it compares behavioral and physiological data in the same animal at the same time points, but there are some concerns that should be addressed before publication.  

Specific comments:

(1) The finding that a-waves were permanently damaged, but b-waves recover is interesting. It is also somewhat counterintuitive since photoreceptors (a-wave) are presynaptic to bipolar cells (b-wave). Can the authors suggest a mechanism of action here that could underlie this result?

(2) ERG traces for the different conditions in Figure 1 should be added.

(3) How was a-wave and b-wave amplitude measured?

(4) In figure 2, the intensity-response curves are given for different post-light damage time points.  Presumably these are based on ERG responses (Figure 1). However, the times given on the x-axes are different for Figures 1 and 2.  Why aren’t the axes the same?  Where do the additional post light exposure time points listed in Figure 2 come from? Please explain.

(5) The authors report that they found greater difference in ERGs as a result of light damage compared to their behavioral results. What impact would this have for the animal, as both are functional measures.

(6) Please indicate where significant differences are located in Figure 5 (with asterisks or something similar).

(7) Methods, lines 231-233 – vendor seems misplaced

(8) Methods, line 233 - How was the stimulating wavelength selected?

(9) Were ERGs recorded before behavior or behavior before ERG? Or was it randomized?

(10) section 4.5,  lines 306-307 states “Only animals surviving 48 days past light, or sham, damage were considered”  How many animals were removed/didn’t make this threshold?  What were the N’s per treatment?  Physiological data suggests damage occurs early (R6), so why not assess retinal anatomy at that time point?  Why was 48 days selected?

(11) line 196 – ‘rats’ is duplicated

(12) line 197 states no “significant functional deficits”. What does this refer to?  Physiological data showed deficits (such as the observed significant change in a-wave) and can be considered functional.  What are the authors referring to?  Or are they referring to a specific time point post light damage?

(13) line 200 states ‘after light damage there was a small decrease in the light-adapted a-wave at R6’ – was it a small change?  It was significantly different.

(14) end of the manuscript, 213-216 – this text could be deleted as it is formatting comments. The authors should replace it with a conclusion.  

(15) The authors should include discussion of how their results may inform other studies that focus on visual behaviors alone to assess the impact of light damage.   What does it mean that a greater response or effect of light damage was seen with ERG compared to behavioral assessments? 

Author Response

Response to Review of IJMS manuscript #156224

The revisions below were extensive, and they greatly improved the manuscript and the telling of what was a complex and thorough set of experiments. This updated manuscript is a better representation of the student’s hard work and the authors thank the reviewers for our elevated demonstration of rigor. We have added two figures and corrected all the minor typos and hopefully all of the confusing language. We have added fourteen new references and rewritten the descriptions of the results and discussion.

Comment from Review #2

  1. Please specify the type of visually guided behavior performed (line 10).
    1. Done
  2. Please specify the duration of the follow-up because the state: “before and after” (line 14) might be misleading suggesting there were just two-time points.
    1. Done
  3. According to the previous point, please describe results like b-wave and photopic behavioral, in terms of trends (for instance: the b-wave amplitude progressively recovered).
    1. Done
  4. For a complete description please state about photopic CFF and scotopic behavior data.
    1. We have added language to describe in better detail the CFF of the two tests,

INTRODUCTION: The introduction is well written however, the state in line 38 which describes this type of light damage as a model of Retinitis Pigmentosa might be misleading because this classic animal model was extensively characterized by literature and used as a general model of retinopathy. Accordingly, there are numerous studies that confirm Cicerone et al. data 1,2,3 therefore the state about inconsistency needs to be briefly assessed (line46). I would kindly suggest to introduce the physiological meaning of CFF in line 47.

            We have expanded the description of these measures with additional references cited, and added the suggested references – thank you for this help.

RESULTS: This section is straightforward and clear. The authors explained the section based on the results obtained and categorized them. However, the following needs attention.

  1. Please define Rmax (line 62).  Done
  2. As this in-vivo model of retinal neurodegeneration can be modulated according to time and light exposure, it might be slightly different from others used in literature. Therefore, it might be important to characterize it by providing statistical tests in every trend description like:
  • B-wave recovered over 80 % (Line 67)
  • A-wave was permanent (line 68)
  • 33% loss (line 73)
  • Progressively declining… (line 82)
  • Decline by 36% (line 84)
  • Decline of 30% (line 85)
  • Etc… we have indicated Anova analyses where completed.
  1. According to the 2nd point, please provide statistical markers in figure 1.
    1. Statistical markers (p<0.05) for individual time points are indicated with an asterisk, the trend analyses done by ANOVA is harder to represent graphically.

  1. Please provide a better definition or reference about I1/2, does the sentence in line 87 miss something?
  2. Please provide statistical markers in figure 2.   One has been added.
  3. Please provide IR unit (line 95) It has been added.
  4. ERG CFF values in figure 3c do not seem “generally higher” than Behavior CFF. Please explain or revise the sentence in line 101. According to these graphs, Hz described in lines 114-115 seem inverted.
    1. Apologies this Figure was an early draft where the photopic ERG and Behavior data are reversed, that is in the wrong columns.  We have remade the figure. And apologize for the confusion.
  5. Please insert the legend and statistical markers in figure 3.
  6. Interesting statistical differences described in lines 103-108 and 117-120 were not shown in figure 3. I would suggest adding representative graphs bar which compare behavior and ERG CFF.  (Not sure if you mean an additional figure 0r supplementary figure
  7. Despite the most represented photoreceptors in a rat retina are rods, the staining in figure 4 does not selectively discriminate just this type of cell, please revise line 122.
    1. The OS layer as defined by the distance between the outer edge of the ONL and the RPE layer is indeed narrower/shrunken. We have added explanatory text.
  8. Please state the type of labeling in line 136. We have added “hemotoxylin”
  9. Line 128 and y axes of figure 5, please revise the unit. The units of micrometers (µm) is correct.
  10. Lines 103/127/128 please add relative standard error values.
  11. Please add which time point (R6/R20/R48/R90) figure 5 refers.   We have added this information (end of study R90)
  12. The ONL area measurements might be misleading because neuroinflammatory processes might produce edema1. I would suggest an additional analysis of photoreceptors rows or ratios about ONL thickness /total retina thickness.
    1. This type of comparison of ONL thickness vs. ONL nuclei count has been verified previously for light damage in rats. We have added a comment and reference,

DISCUSSION and CONCLUSION: The discussion is quite clear but the following suggestions need to be addressed.

  1. According to the 2nd point, please revise the discussion selecting just statistically significant differences (line 147-152). Furthermore, based on this particular animal model, comparisons among different studies require taking into account all variables involved. As instance, in Willams experiment rats were breaded at 5 lux and exposed to 500lux (100 times more than usual) on the other hand your study light damage exposure was just double. Please revise 194-199 lines.
  2. Do the authors might consider whether differences between ERG CFF and behavioral CFF might be related to different technique properties? Like anesthesia, pupil dilatation, anxiety level, body temperature, distances from displays…
    1. We have added comments in the discussion.
  3. Please complete references with the most updated studies about long-term apoptosis followed by light exposure (line 205).   We have added references and discussion.
  4. 213-216 lines should be removed.   Done
  5. The discussion lacks in final perspectives, please revise it describing possible step forwards led by your study.
    1. We have added this commentary as requested.

MATERIALS AND METHODS:

  1. Please consider that retinal damage by light exposure is not homogeneous, the superior retina is much more sensitive to light than the inferior, do the author consider the eye orientation while they made slides?
    1. Indeed we maintained the superior/inferior axis identification and we find no regional difference in rod loss. (for cone loss see companion paper) Clark et al. We have added commentary in the discussion about differences in our results compared to other studies.
  2. Is the distance of liquid crystal shutter in the ERG equivalent to the displays in the behavioral test?
    1. These are unrelated as the ERG illumination is funneled through a fiber optic after all filtering. The liquid crystal shutter is an optical device that produces sinusoidal modulation of the light prior to it being focused upon the fiber optic. The light projected upon the rear of the behavioral display was a separate device, an electronically controlled LED light.
  3. The authors used parametric tests for statistics, were they preceded by normality test?
    1. No, it was assumed and we have added this comment
  4. I would suggest to add a scheme about behavioral test.
    1. The behavioral training and testing is described in detail and we have added of schematic image of the apparatus in Fig S2.

References

  1. Riccitelli S, Di Paolo M, Ashley J, Bisti S, Di Marco S. The Timecourses of Functional, Morphological, and Molecular Changes Triggered by Light Exposure in Sprague–Dawley Rat Retinas. Cells. 2021;10(6):1561. doi:10.3390/cells10061561
  2. Polosa A, Liu W, Lachapelle P. Retinotopic distribution of structural and functional damages following bright light exposure of juvenile rats. PLoS One. 2016;11(1):1-20. doi:10.1371/journal.pone.0146979
  3. Sugawara T, Sieving PA, Bush RA. Quantitative relationship of the scotopic and photopic ERG to photoreceptor cell loss in light damaged rats. Exp Eye Res. 2000;70(5):693-705. doi:10.1006/exer.2000.0842

Reviewer #3 comments

The aim of the present research project is to compare physiological (ERG) and behavioral (water maze) measures after inducing mild retinal degenerative process to define whether the two methods provide similar or different information. The question is relevant but there are several critical points raised by the experimental protocol, presentation of data and discussion.

The intensity of damaging light is relatively low (280 lux) also considering the mean luminance in animal room (between 57-140 lux) but 10 days of continuous exposure might alter circadian rhythm and modulate initial results this point has to be discussed.  Discussion added.

ERG was recorded in dark and light condition it could be important to present original records and an intensity response curve to show how sensitivity was calculated.

A new Figure 1 has been added.

The same observation for the CFF, was it performed an FFT analysis of the response? Was it taken as a response the second harmonic amplitude as a function of the temporal frequency?

There was no FFT,  rather the method followed that of Prusky et al. to measure the fundamental response amplitude and a linear regression allowed us to identify a threshold criterion response of 3 µV.  Commentary and references have been added.

In water maze experiments it would be convenient to present a learning curve. It is not clear what the authors mean with two luminance level used.

The two luminance levels were chosen to isolate the rod- and cone-driven retinal responses. We have added a statement to clarify.

An additional observation comes from histological results, it has been widely reported that light damage induced a localized damage on the dorsal area named “hot spot” here we have a uniform loss of photoreceptors how do the authors explain this result? There is a recent paper (Riccitelli et al. cells 2021) reporting functional and morphological data on Light Damaged Sprague–Dawley and related literature it might be important to include it in the discussion. The discussion is not well organized and conclusions remain unclear.

We have rewritten the discussion to encompass more recent results. Our long period of elevated light levels did not produce a hot-spot of damage suggesting that animal movement over the 10-day period was either less inhibited by the light levels, or that normal feeding/socialization behavior resulted in a more uniform illumination of the retina and  thus more uniform damage; we have added commentary to the discussion.

Reviewer #1

In this manuscript, the authors describe their experiments correlating ERG CFF and behavioral CFF after light-damage. They expose albino rats to 10 days of low light intensity damage and then assess ERGs and behavior 6, 20, 48, and 90 days after exposure. Their results indicate that light-damage reduced ERG amplitude at R6, with deficits in both a-wave and b-wave amplitudes.  A-wave amplitudes displayed a permanent reduction, while b-wave amplitude displayed recovery at the later time points.  The authors also report differences in light-damage effect between ERG CFF and behavioral CFF, with greater changes in ERG responses.  This work is relevant as it compares behavioral and physiological data in the same animal at the same time points, but there are some concerns that should be addressed before publication.  

Specific comments:

(1) The finding that a-waves were permanently damaged, but b-waves recover is interesting. It is also somewhat counterintuitive since photoreceptors (a-wave) are presynaptic to bipolar cells (b-wave). Can the authors suggest a mechanism of action here that could underlie this result?

            We have added commentary and  references in the discussion section.

(2) ERG traces for the different conditions in Figure 1 should be added.

            We have added a new figure 1 giving these sample data.

(3) How was a-wave and b-wave amplitude measured?

            We have added a description and additional reference to the Electroretinography Methods.

(4) In figure 2, the intensity-response curves are given for different post-light damage time points.  Presumably these are based on ERG responses (Figure 1). However, the times given on the x-axes are different for Figures 1 and 2.  Why aren’t the axes the same?  Where do the additional post light exposure time points listed in Figure 2 come from? Please explain.

The ERG measures in Figure 1 and  Figure 2 came from different cohorts of animals, and given the difficulty of experiments the second cohort was terminated earlier after sensitivity recovery had stabilized, perhaps after adjustments due to cell debris removal and possible synaptic remodeling were complete.

(5) The authors report that they found greater difference in ERGs as a result of light damage compared to their behavioral results. What impact would this have for the animal, as both are functional measures.

The ERG flash responses are measures of maximal voltage and current generated by the entire retina, whereas flicker threshold measures are signal to noise measures at the very edge of detection.  We will add this to our discussion.

(6) Please indicate where significant differences are located in Figure 5 (with asterisks or something similar).

            A point by point comparison is not valid as all the point need to be considered simultaneously – thus and area calculation was made

(7) Methods, lines 231-233 – vendor seems misplaced  Fixed

(8) Methods, line 233 - How was the stimulating wavelength selected?  According to the optimum for rat rhodopsin -we’ve added the reference.

(9) Were ERGs recorded before behavior or behavior before ERG? Or was it randomized?\

            ERG were recorded after behavioral testing to eliminate any possible lingering effects of anesthesia. We’ve added this note to the methods section.

(10) section 4.5,  lines 306-307 states “Only animals surviving 48 days past light, or sham, damage were considered”  How many animals were removed/didn’t make this threshold?  What were the N’s per treatment?  Physiological data suggests damage occurs early (R6), so why not assess retinal anatomy at that time point?  Why was 48 days selected?

The n values are given for each figure in the legends. Bright light exposure can produce massive damage to the retina, but it can take more than a week for the damaged and dying cells to be removed, and potential  remodeling of synaptic confounds the physiological measures.

(11) line 196 – ‘rats’ is duplicated  Fixed

(12) line 197 states no “significant functional deficits”. What does this refer to?  Behvioral CFF, we have added language to clarify.

Physiological data showed deficits (such as the observed significant change in a-wave) and can be considered functional.  What are the authors referring to?  Or are they referring to a specific time point post light damage? We have added CFF to clarify.

(13) line 200 states ‘after light damage there was a small decrease in the light-adapted a-wave at R6’ – was it a small change?  It was significantly different.

We stated the small change because the trend is in that direction and ultimately (at 90 days. The photopic a-wave was statistically reduced.

(14) end of the manuscript, 213-216 – this text could be deleted as it is formatting comments. The authors should replace it with a conclusion.  We have expanded the conclusion section.

(15) The authors should include discussion of how their results may inform other studies that focus on visual behaviors alone to assess the impact of light damage.   What does it mean that a greater response or effect of light damage was seen with ERG compared to behavioral assessments? 

We have expanded the discussion to elaborate on maximal retinal voltages, vs. threshold measure of flicker (Behavioral or ERG).

Reviewer 2 Report

Comments and Suggestions for Authors

Dear authors,

The article titled ‘ERG and Behavioral CFF in Light-Damaged Albino Rats’ submitted by Rubin et al. to Molecular Science journal investigates correlations between flicker ERG and behavioral responses in albino rats exposed to

MINOR AND MAJOR CORRECTIONS NEEDED:

TITLE: The title properly pointed out the article's topic.

ABSTRACT: The abstract is comprehensive and clear. Essential information relevant to the findings of the study is incorporated. However, the authors should add more details and clarify the following points:

  1. Please specify the type of visually guided behavior performed (line 10).
  2. Please specify the duration of the follow-up because the state: “before and after” (line 14) might be misleading suggesting there were just two-time points.
  3. According to the previous point, please describe results like b-wave and photopic behavioral, in terms of trends (for instance: the b-wave amplitude progressively recovered).
  4. For a complete description please state about photopic CFF and scotopic behavior data.

INTRODUCTION: The introduction is well written however, the state in line 38 which describes this type of light damage as a model of Retinitis Pigmentosa might be misleading because this classic animal model was extensively characterized by literature and used as a general model of retinopathy. Accordingly, there are numerous studies that confirm Cicerone et al. data 1,2,3 therefore the state about inconsistency needs to be briefly assessed (line46). I would kindly suggest to introduce the physiological meaning of CFF in line 47.

RESULTS: This section is straightforward and clear. The authors explained the section based on the results obtained and categorized them. However, the following needs attention.

  1. Please define Rmax (line 62).
  2. As this in-vivo model of retinal neurodegeneration can be modulated according to time and light exposure, it might be slightly different from others used in literature. Therefore, it might be important to characterize it by providing statistical tests in every trend description like:
  • B-wave recovered over 80 % (Line 67)
  • A-wave was permanent (line 68)
  • 33% loss (line 73)
  • Progressively declining… (line 82)
  • Decline by 36% (line 84)
  • Decline of 30% (line 85)
  • Etc…
  1. According to the 2nd point, please provide statistical markers in figure 1.
  2. Please provide a better definition or reference about I1/2, does the sentence in line 87 miss something?
  3. Please provide statistical markers in figure 2.
  4. Please provide IR unit (line 95)
  5. ERG CFF values in figure 3c do not seem “generally higher” than Behavior CFF. Please explain or revise the sentence in line 101. According to these graphs, Hz described in lines 114-115 seem inverted.
  6. Please insert the legend and statistical markers in figure 3.
  7. Interesting statistical differences described in lines 103-108 and 117-120 were not shown in figure 3. I would suggest adding representative graphs bar which compare behavior and ERG CFF.
  8. Despite the most represented photoreceptors in a rat retina are rods, the staining in figure 4 does not selectively discriminate just this type of cell, please revise line 122.
  9. Please state the type of labeling in line 136.
  10. Line 128 and y axes of figure 5, please revise the unit.
  11. Lines 103/127/128 please add relative standard error values.
  12. Please add which time point (R6/R20/R48/R90) figure 5 refers.
  13. The ONL area measurements might be misleading because neuroinflammatory processes might produce edema1. I would suggest an additional analysis of photoreceptors rows or ratios about ONL thickness /total retina thickness

DISCUSSION and CONCLUSION: The discussion is quite clear but the following suggestions need to be addressed.

  1. According to the 2nd point, please revise the discussion selecting just statistically significant differences (line 147-152). Furthermore, based on this particular animal model, comparisons among different studies require taking into account all variables involved. As instance, in Willams experiment rats were breaded at 5 lux and exposed to 500lux (100 times more than usual) on the other hand your study light damage exposure was just double. Please revise 194-199 lines.
  2. Do the authors might consider whether differences between ERG CFF and behavioral CFF might be related to different technique properties? Like anesthesia, pupil dilatation, anxiety level, body temperature, distances from displays…
  3. Please complete references with the most updated studies about long-term apoptosis followed by light exposure (line 205).
  4. 213-216 lines should be removed.
  5. The discussion lacks in final perspectives, please revise it describing possible step forwards led by your study.

MATERIALS AND METHODS:

  1. Please consider that retinal damage by light exposure is not homogeneous, the superior retina is much more sensitive to light than the inferior, do the author consider the eye orientation while they made slides?
  2. Is the distance of liquid crystal shutter in the ERG equivalent to the displays in the behavioral test?
  3. The authors used parametric tests for statistics, were they preceded by normality test?
  4. I would suggest to add a scheme about behavioral test.

References

  1. Riccitelli S, Di Paolo M, Ashley J, Bisti S, Di Marco S. The Timecourses of Functional, Morphological, and Molecular Changes Triggered by Light Exposure in Sprague–Dawley Rat Retinas. Cells. 2021;10(6):1561. doi:10.3390/cells10061561
  2. Polosa A, Liu W, Lachapelle P. Retinotopic distribution of structural and functional damages following bright light exposure of juvenile rats. PLoS One. 2016;11(1):1-20. doi:10.1371/journal.pone.0146979
  3. Sugawara T, Sieving PA, Bush RA. Quantitative relationship of the scotopic and photopic ERG to photoreceptor cell loss in light damaged rats. Exp Eye Res. 2000;70(5):693-705. doi:10.1006/exer.2000.0842

Author Response

(The authors gave the same response as above.)

Reviewer 3 Report

The aim of the present research project is to compare physiological (ERG) and behavioral (water maze) measures after inducing mild retinal degenerative process to define whether the two methods provide similar or different information. The question is relevant but there are several critical points raised by the experimental protocol, presentation of data and discussion.

The intensity of damaging light is relatively low (280 lux) also considering the mean luminance in animal room (between 57-140 lux) but 10 days of continuous exposure might alter circadian rhythm and modulate initial results this point has to be discussed.

ERG was recorded in dark and light condition it could be important to present original records and an intensity response curve to show how sensitivity was calculated.

The same observation for the CFF, was it performed an FFT analysis of the response? Was it taken as a response the second harmonic amplitude as a function of the temporal frequency?

In water maze experiments it would be convenient to present a learning curve. It is not clear what the authors mean with two luminance level used.

An additional observation comes from histological results, it has been widely reported that light damage induced a localized damage on the dorsal area named “hot spot” here we have a uniform loss of photoreceptors how do the authors explain this result? There is a recent paper (Riccitelli et al. cells 2021) reporting functional and morphological data on Light Damaged Sprague–Dawley and related literature it might be important to include it in the discussion. The discussion is not well organized and conclusions remain unclear.

Author Response

(The authors gave the same response as above.)

Round 2

Reviewer 1 Report

The authors have addressed all concerns of this reviewer. Thank you.

Author Response

We thank the reviewer for the corrective advice.

Reviewer 3 Report

In the revised version of the paper the authors nicely clarified the majority of the questions raised and the paper was improved. One point deserves attention and it is the one related to the absence of the “hot spot” I don’t think is related to uniform retinal illumination but it seems more subtle and /or threshold problem in any case in mice no matter the intensity used it was never noticed a sensitive dorsal area and several authors (Valter k. for example) discussed of dorsal retinal specialization in rats. Probably it is better to say that there is no simple explanation. A second point is the recovery of the b-wave it has been reported activation of neuroprotective mechanism which controlled transfer of visual information from the first to the second retinal neurons and probably even more complex mechanisms this might explain the relation low level damage and increased retinal resilience.

Author Response

In the revised version of the paper the authors nicely clarified the majority of the questions raised and the paper was improved.

Thank you acknowledging the improvements.

(1) One point deserves attention and it is the one related to the absence of the “hot spot” I don’t think is related to uniform retinal illumination but it seems more subtle and /or threshold problem in any case in mice no matter the intensity used it was never noticed a sensitive dorsal area and several authors (Valter k. for example) discussed of dorsal retinal specialization in rats. Probably it is better to say that there is no simple explanation.

We have added references and discussion to address the complexity of spatial inhomogeneity of retinal damage and recovery.

(2) A second point is the recovery of the b-wave it has been reported activation of neuroprotective mechanism which controlled transfer of visual information from the first to the second retinal neurons and probably even more complex mechanisms this might explain the relation low level damage and increased retinal resilience.

We have added references and discussion points to note the reactive neuroprotective mechanisms discovered in other studies.